

# Cloud Tolerance of Remote Sensing Technologies to Measure Land Surface Temperature

Thomas R.H. Holmes[12], Christopher Hain[3], Martha C. Anderson[1], Wade T. Crow[1]

[1]Hydrology and Remote Sensing Lab., USDA-ARS, Beltsville, MD, USA
[2]Science Systems and Applications, Greenbelt, MD, USA
[3]Earth Science Interdisciplinary Center, University of Maryland, College Park, MD, USA

*Correspondence to*: Thomas R.H. Holmes (thomas.holmes@ars.usda.gov)

**Abstract.** Conventional means to estimate land surface temperature (LST) from space relies on the thermal infrared (TIR) spectral window and is limited to cloud-free scenes. To also provide LST estimates during periods with clouds, a new
method was developed to estimate LST based on passive microwave (MW) observations. The MW-LST product is informed by 6 polar orbiting satellites to create a global record with up to 8 observations per day for each 0.25° resolution grid box. For days with sufficient observations, a continuous diurnal temperature cycle (DTC) was fitted. The main characteristics of the DTC were scaled to match that of a geostationary based TIR-LST product.

This paper tests the cloud tolerance of the MW-LST product. In particular, we demonstrate its stable performance with
respect to flux tower observation sites (4 in Europe and 9 in the United States), over a range of cloudiness conditions up to heavily overcast skies. The results show that TIR-based LST has slightly better performance than MW-LST for clear sky observations but suffers an increasing negative bias as cloud cover increases. This negative bias is caused by incomplete masking of cloud covered areas within the TIR scene that affects many applications of TIR-LST. In contrast, for MW-LST we find no direct impact of clouds on its accuracy and bias. MW-LST can therefore be used to improve TIR cloud screening.
Moreover, the ability to provide LST estimates for cloud-covered surfaces can help expand current clear-sky-only satellite retrieval products to all-weather applications.

## 1 Introduction

Information about the temperature of the land surface (LST) is an important element in the retrieval of many hydrological states and fluxes from satellite measured radiances. For example, the retrieval of soil moisture or precipitation from passive
microwave observations requires a coincident estimate of LST (e.g. Owe et al., 2008). In other applications, the rate of change in temperature is contrasted with net radiation to estimate evaporation as a residual of the surface energy balance (e.g. Anderson et al., 2011).

The most direct way to estimate LST from space-borne instruments is by radiometers which measure within the thermal band of the infrared spectrum (TIR). Thermal emission within this frequency band can be related directly to the physical
temperature of the land surface, and is more precisely termed the ensemble radiometric temperature (Norman and Becker,





1995). Space-borne TIR radiometers allow for very high spatial resolution imagery. Even at the height of geostationary platforms, some radiometers deliver 2-km spatial resolution. A drawback to the TIR technique is that - at such wavelengths - clouds completely block the emission from the land surface. This means that space-borne TIR radiometers give no information about the land surface below the clouds, and instead reflect the temperature and emissivity of the clouds. The

result is that the quality of the cloud screening directly affects the quality of a TIR LST product.

An alternative, more cloud-tolerant technique is based instead on passive microwave (MW) observations. In particular the Ka-band (~37 GHz) was shown to have a strong link with LST (Owe and Van de Griend, 2001). Based on these observations, linear regression-based LST estimates were derived for the Ka-band (Holmes et al., 2009) and variants of these linear relations are currently used in soil moisture retrieval (Jackson et al., 1999; Owe et al., 2008). However, using a single

linear regression across the globe ignores potentially significant differences in microwave emissivity and can result in large biases, especially in desert areas. It also cannot account for large differences in the amplitude of the diurnal cycle between MW- and TIR-based LST, which have been implicated in reduced soil moisture retrieval skill during daylight hours (Lei et al., 2015; Parinussa et al., 2011). In contrast to these relatively simple linear methods, a neural network method was developed by Aires et al. (2001) to estimate LST based on multiple microwave channels besides Ka-band. By using

atmospheric and surface information in addition to TIR LST in the training of the scheme, this method minimized systematic bias in monthly mean temperatures. However, because the training is based on a single polar orbiting satellite, it cannot give diurnal temperature information. Both of these methods were compared with station data by Catherinot et al. (2011), giving strong confirmation on the lack of sensitivity of microwave-based LST estimates to cloud liquid water path. One drawback to all passive microwave-based methods is relatively coarse spatial resolution as compared to TIR sensors. At Ka-band, the

smallest footprint size currently achieved with polar orbiting satellites is 10 x 15 km.

Because of the complementary nature of TIR and MW-based LST, there is a clear interest in merging these two independent technologies. For example, TIR-LST based evaporation retrievals would benefit from observational data during cloudy periods (e.g. Anderson et al., 2011). On the other hand, microwave soil moisture retrievals from SMAP have the goal of 9 km spatial resolution and this poses a resolution challenge to MW-LST inputs if TIR-LST cannot be leveraged. As a first

step towards an ultimate merger of diurnally continuous TIR and MW LST products, Holmes et al. (2015) developed a method to scale the diurnal characteristics of multi-satellite dataset of Ka-band observations to TIR-LST. Because well validated TIR-LST products exist from geostationary satellites, this scaling was able to account for biases in characteristics of the diurnal temperature cycle (DTC). By explicitly taking account of systematic differences in DTC between TIR and Ka-band, this method is able to estimate LST at any time of day.

The aim of this paper is to evaluate the new global MW-LST dataset in comparison to existing TIR-LST data over clear-sky days and to test the assumption that MW-LST is tolerant to high levels of cloud coverage. Ground observations provide a common benchmark to test the relative accuracy of the two satellite products. Because the diurnal MW-LST product (Holmes et al., 2015) was scaled to TIR-based LST as produced by the Land Surface Analysis Satellite Application Facility (LSA-SAF, see http:/landsaf.meteo.pt), the evaluation is mostly concerned with temporal precision, not with absolute bias.





In previous work it was shown that relative aspects of a coarse-scale product can be evaluated using sparse in situ observations (Holmes et al., 2012). For a thorough discussion of absolute accuracy, readers are referred to papers detailing validation exercises for LSA-SAF-LST (Ermida et al., 2014; eg. Göttsche et al., 2013).

After establishing the accuracy of MW-LST relative to TIR-LST for a particular site, the stability of the precision of MW-
LST (relative to ground data) for increasing cloudiness will be tested. Previous work showed indications of cloud tolerance of MW-LST in comparison to TIR-LST (Holmes et al., 2015), but the analysis used proxies for both cloud cover and LST quality. In this paper we use a more direct estimate of cloudiness and provide a more detailed look at the validation statistics for different levels of daytime cloudiness with the ground station as the reference. The hypothesis we test is that clouds affect a satellite measured LST by introducing an error (E). If E is consistent in sign throughout the measurement period (e.g.
if clouds always lower the satellite LST estimate) this will introduce a systematic bias that will increase with cloud cover. If on the other hand the sign of E varies it will increase the random error in LST, but not necessarily a systematic bias. Only if we do not see a systematic bias with increasing cloud cover, nor an increase in random error can we reject the hypothesis that clouds affect the satellite LST.

## 2. Materials

### 2.1 Satellite LST estimates: Thermal infrared

TIR-LST is available from many sources, including both low-earth orbiting satellites and geostationary satellites. Because of our interest in the diurnal features of LST, used in surface energy balance evaluations, this study focuses on TIR-LST products developed from geostationary satellites. The first product is based on the Spinning Enhanced Visible and Infrared
Imager (SEVIRI) onboard the Meteosat Second Generation (MSG-9) satellite. MSG-9 is positioned over the equator at 0° longitude. It has geographic coverage of Africa, Europe and the east coast of South America (with incidence angles below 70°). The Land Surface Analysis, Satellite Application Facility (LSA SAF) produces operational LST products based on split window observations (channels centered at 10.8 μm and 12.0 μm) of MSG-9. LSA SAF LST is originally produced at 3 km spatial resolution. For this study, the data is aggregated to match the 0.25° resolution of MW-LST.

For North-America, NOAA operates the Geostationary Operational Environmental Satellites (GOES). GOES Surface and Insulation Products (GSIP) are produced by the office of Satellite Data Processing and Distribution, NESDIS, NOAA, and includes LST (V3 was used for this study). Unlike MSG-based LST, GOES GSIP LST is based on a dual window technique (3.9 and 11.0 μm), rather than the preferred split window technique due to the lack of a 12 micron thermal channel on the current generation of the GOES imager. An operational hourly LST at 0.125° degree spatial resolution is available from 2
April 2009 onwards. For this study, we averaged the nine 0.125° degree nodes that cover the edge and center of the 0.25° MW-LST product grid cell. In order to further reduce possible cloud contamination, a particular data point is only used if all nine 0.125° degree pixels covering the 0.25° grid box contain (non-cloud-masked) data.





### 2.2 Satellite LST estimates: Microwave

The MW-LST product is based on vertical polarized Ka-band (36-37 GHz) brightness temperature ($T^{Ka}$) as measured by microwave radiometers on 6 satellites in low earth orbit. These observations are combined to create a global record with up to 8 observations per day for each 0.25° resolution grid box. The data are binned in 15 minute windows of local solar time

(0:00-0:15 is first window of the day), and inter-calibrated using observations from the TRMM satellite (with an equatorial overpass) as a transfer reference. Individual 0.25° averages are masked if the spatial standard deviation of the oversampled Ka-band observations exceeds a prior determined threshold for a given grid box. Both the inter-calibration and quality control procedures are described in detail in Holmes et al (2013a).

The methodology to estimate LST from this record of Ka-band observations is described in (Holmes et al., 2015) and

summarized below. For days with suitable observations (a minimum of 4, including at least one within a third day length from solar noon) and no $T^{Ka} < 250\,K$ (an indication of frozen soil), a continuous diurnal temperature cycle (DTC) is fitted. The DTC model used is based on Göttsche and Olesen (2001) with slight adaptations to limit the number of parameters. This implementation (DTC3) is fully described in Holmes et al (2015). DTC3 summarizes the DTC with two daily parameters (daily minimum $T_0$ at start and end of day, and diurnal amplitude $A$) together with diurnal timing ($\varphi$), that is assumed a

temporal constant (Holmes et al., 2013b). The daily mean is defined by the daily minimum and the amplitude ($\bar{T} = T_0 + A\,/2$). The Ka-band DTC parameters for individual days ($\bar{T}_d^{Ka}$, $A_d^{Ka}$) are scaled to match the long term mean of TIR observations:

$$A_d^{MW} = A_d^{Ka}/\delta \qquad\qquad\qquad\qquad\qquad\qquad\qquad\qquad\qquad\qquad (1)$$

$$\bar{T}_d^{MW} = \beta_0 + \beta_1 \bar{T}_d^{Ka} \qquad\qquad\qquad\qquad\qquad\qquad\qquad\qquad\qquad (2)$$

The scaled parameters are indicated with the superscript 'MW'. The parameter $\delta$ represents the slope of the zero-order least squares regression line for estimating the amplitude of TIR-LST ($A_d^{TIR}$) from $A_d^{Ka}$. The intercept ($\beta_0$) and slope ($\beta_1$) to correct the mean daily temperature ($\bar{T}_d^{Ka}$) for systematic differences with TIR-LST ($\bar{T}_d^{TIR}$) are determined with a constrained numerical solver, as in Holmes et al. (2015). The constraint is based on radiative transfer considerations and assures that the scaling of the mean is in agreement with the prior scaling of the amplitude (Eq. 1). These scaling parameters were

determined for each 0.25° grid box based on data for the period 2009-2012. The scaling (Eqs. 1 and 2) is applied to every day for which estimates of $\bar{T}^{Ka}$ and $A^{Ka}$ are available. Together with the timing of the diurnal cycle of TIR-LST, $\varphi^{TIR}$, as determined based on (Holmes et al., 2013b), we then calculate the diurnal MW-LST based on the same DTC3 model:

$$\text{MW-LST} = DTC3(\varphi^{TIR}, T_0^{MW}, A^{MW}) \qquad\qquad\qquad\qquad\qquad\qquad\qquad (3)$$


Comparing actual Ka-band observations to estimates provided by the fitted DTC model provides a valuable means of quality control. The root mean square error (RMSE) of the misfit between the DTC3 model and the sparse $T^{Ka}$ observations is used





to flag days where the assumptions imposed by the shape of clear sky DTC are not valid or individual Ka-band observations have a large bias.

Besides the continuous MW-LST product, we can also evaluate the product at the actual Ka-band observation times (thus weakening our reliance on the DTC3 model). To do this we project the difference between the original MW data and the
DTC model fit onto MW-LST. This product is referred to as MW –LST-Sparse:

$$\text{MW-LST-Sparse} = DTC3(\varphi^{TIR}, T_0^{MW}, A^{MW}) + \frac{\langle A^{TIR} \rangle}{\langle A^{Ka} \rangle}\left(T^{Ka} - DTC3(\varphi^{Ka}, T_0^{Ka}, A^{Ka})\right) \tag{4}$$

**2.3 Ground observations**

FLUXNET is a worldwide network of meteorological measurement towers (flux tower) with common measurement
protocols (Baldocchi et al., 2001). Each flux tower includes an instrument positioned above the vegetation canopy to measure net radiation. This instrument is made up of two pyranometers to measure up and down welling short wave radiation and two pyrgeometers to measure up and down welling long wave radiation. The radiometric surface temperature ($T$) can be derived from the long wave radiation measurements (upwelling: $L^{\uparrow}$, Wm$^{-2}$, and downwelling: $L^{\downarrow}$, Wm$^{-2}$) using the following equation:

$$L^{\uparrow} = \varepsilon_L \sigma T^4 + (1 - \varepsilon_L)L^{\downarrow} \tag{5}$$

where $\varepsilon_L$ is the broadband emissivity over the spectral range of the pyrgeometer (4.5-42 μm), $\sigma$ is the Stefan-Boltzmann constant ($\sigma$=5.670 x 10$^{-8}$ Wm$^{-2}$K$^{-1}$).

**2.3.1 Long wave emissivity**

An estimate of $\varepsilon_L$ is obtained for each site based on measurements of $L^{\uparrow}$ together with additional measurements of screen level air temperature ($T_a$), sensible heat flux ($H$) and wind speed ($u$). This estimate is based on three assumptions: 1) $H$ is directly proportional to the near surface temperature gradient, 2) the difference $T - T_a$ represents that temperature gradient, and 3) $H = 0$ when there is no vertical temperature gradient (i.e. $T = T_a$). With these three assumptions in mind we then
iterate over $\varepsilon_L$ to find the solution where the regression of $H$ against ($T$-$T_a$) goes through zero and the squared errors are minimized. When measurements of wind speed are available they are used to select atmospheric conditions where the relationship between $H$ and the near surface temperature gradient is strongest ($u$> 2 m/s). For forest sites, the direct relationship between $H$ and $T$ gradients breaks down. In those cases the simpler assumption is used that the long term average of $T$ and $T_a$ are equal: $\langle T \rangle = \langle T_a \rangle$. For more discussion and examples of this method see Holmes et al. (2009). In
this study we apply this method to determine monthly $\varepsilon_L$ for each site individually, and then use the median value of $\varepsilon_L$



(listed in Table 1) to calculate $T$ based on Eq. 5. The standard deviation of the monthly measurements of $\varepsilon_L$ are also listed in Table 1 and provides an indication of both uncertainty and seasonal variation in $\varepsilon_L$ .

**2.3.2 Spatial representativeness and selection tower sites**

The tower-based estimate of $T$, from (Eq. 5), directly represents only the immediate tower surroundings within a radius of
approximately 50 m. Clearly this is a very small spatial sampling of the 0.25° grid box (~25x25 km) represented by the satellite LST estimates used in this study (see Section 2.1 and 2.2). As a consequence, we typically find large systemic differences between the station data and the areal average. Given that overall weather conditions are relatively homogenous over distances of 25 km, these differences can be attributed to the land cover type at the station location in comparison to that over the entire grid box (for examples of this, see Holmes et al. (2009)). The representation of the spatial average by
ground observations can be improved significantly if more than one station is available in the same grid box and the towers are situated in thermodynamically contrasting land cover types: (forest and cropland/herbaceous). In that case the land cover associated with the tower site(subscript, s) determines the weight ($W$) according to the spatial fraction of that land cover type within the 0.25° grid box (MCD12C1, Friedl et al. (2010)). We use this information to estimate the grid average LST as the weighted average of site measured $T$ according to Eq. (6):

$$LST = \frac{\sum_{s=1}^{n} W_s T_s}{\sum_{s=1}^{n} W_s}. \tag{6}$$

For example, site DE-Hai of location A is located in a forest and represents 16 % of the pixel. Site DE-Geb is located in croplands and represents the 80 % of the pixel that has low vegetation cover or bare soil. Urban, open water, or wetland account for the remaining 4 %, which does not affect the weighting. We only considered locations where this rest fraction is below 5 % of the grid coverage. Another criterion for site selection was that the land cover at the site must represent more
than 75 % of the pixel. Sites in mountainous areas are also excluded. For the period of 2009-2012 this means there are 13 locations with at least 2 years of flux tower sites available for this study, and four of these locations contain multiple stations. For locations where only one stations is available, LST is set equal to the site measurement: $LST = T$. All the validation targets are listed in Table 1, together with the geographic location of the individual stations, the land cover type as reported by the flux tower operators and the parameters $W$ and $\varepsilon_L$ as described above.

**2.3.3 Cloudiness at tower location**

The down welling short wave radiation ($S^{\downarrow}$) as measured at the fluxtower is strongly affected by the amount of condensed water in the atmosphere. We can therefore use the reduction in site measured daytime $S^{\downarrow}$ relative to an expected value during clear skies as a proxy for cloudiness. The clear-sky irradiance $S^{\downarrow}_{clear}$ is estimated based on top-of-atmosphere solar irradiance ($S^{TOA}$) which can be calculated based on geographic location and day of year (Van Wijk and Ubing, 1963). Even on a clear
day, atmospheric absorption reduces the irradiance at the surface by 20 to 30 % from the top of atmosphere value. We





estimate this clear sky absorption ($A_{clr}$) by calculating the slope ($\beta$) of the zero order linear regression between $S^{\downarrow}$ and $S^{TOA}$ for days that are in the highest quintile of $S^{\downarrow}/S^{TOA}$ : $A_{clr}=1-\beta$. These estimates of $A_{clr}$ (listed in Table 1 for each individual site) range from 0.22 to 0.31 and show a good agreement between stations of the same cluster. We use the minimum recorded value for each cluster to calculate $S^{\downarrow}_{clear}$:

$$S^{\downarrow}_{clear} = S^{TOA}(1 - A_{clr}) . \tag{7}$$

By using $S^{\downarrow}_{clear}$ to normalize measured $S^{\downarrow}$, we account for solar zenith effects and can formulate a measure for shortwave cloud absorption ($A_{cloud}$), expressed in percentage:

$$A_{cloud} = 100 \frac{S^{\downarrow}_{clear}-S^{\downarrow}}{S^{\downarrow}_{clear}} \tag{8}$$

This definition of $A_{cloud}$ is used as a measure of cloudiness and calculated based on 3 hour totals of insolation for the daytime between 6AM to 6PM. Obviously this approach does not work for night-time. For night-time hours we use the neighbouring daytime window:

$$A_{cloud}(0 - 6) = A_{cloud}(6 - 12) \tag{9}$$
$$A_{cloud}(18 - 24) = A_{cloud}(12 - 18) \tag{10}$$

Figure 1 gives an example of the site measured $S^{\downarrow}$ and the calculated $A_{cloud}$ for an 8-day summer period at station B (top panel). The bottom panel shows the site measured LST and illustrates how the temporal sampling of the satellite products is affected by clouds.

## 3. Results

We acquired data for 17 field sites in 13 unique grid locations with data records within the 4-year time period of 2009 to 2012. Table 2 lists the amount of days with at least 12 hours of observations for either MW or TIR-LST to give an overall sense of available validation data for this study . In total we have 13316 data days of in situ data (36 out of 44 data years). Of these data days, 50 % also have MW-LST estimates, and 36 % have TIR-LST observations. For MW, this percentage is negatively affected by the gap between AMSR-E (radiometer turned off on October, 2011) and AMSR2 (first observations in July 2012). The MW-LST product is heavily-reliant on these satellites with a midday overpass for constraining the diurnal amplitude. For TIR, the percentage is particularly low for GOES due to a more stringent cloud filter than employed for MSG.

To better represent the relative data coverage that is possible with the two LST retrieval techniques, we focus on the four station pairs in the MSG domain and limit the time period to 2009-2011. We further focus on the days where the minimum temperature (as measured at the station) stays above freezing (the MW method is not applicable for subfreezing temperatures). Within this smaller subset, we have 2506 data days of in situ data and the coverage of MW is 55 % in comparison with the 42 % coverage for TIR. However, breaking this down by cloud cover reveals the big difference in




coverage resulting from the wavelength-dependant tolerance to clouds. During clear skies, the coverage of TIR is 93 %, and MW comes in at 76 % (mainly attributed to the 2/3 days coverage for AMSR-E). During cloudy days the coverage drops to 13 % for TIR, whereas for MW it maintains 55 % coverage.

In the following section we want to answer two questions. How does the MW LST compare to TIR LST in relation to ground data during days with clear skies? And is the performance of MW-LST affected by clouds? We focus on hourly average temperatures for days where the station data remains above 1°C to avoid snow or frozen surface conditions.

### 3.1 Clear-sky comparison of satellite LST products

The ground observations provide a common benchmark to test the relative accuracy and bias of the two satellite products with the same in situ data. Days with clear skies are selected based on the measure of cloudiness as defined in Section 2.3.3, with a maximum accepted value of $A_{cloud} = 0.2$. This is in addition to the cloud screening performed in the generation of the TIR products (Section 2.1), the quality control of the MW LST (Section 2.2), and the selection of frost-free days. Even though the spatial representativeness and uncertainty in $\varepsilon_L$ may insert systematic errors in the estimation of the spatial average from the ground stations, they can be used as a reference to compare different satellite LST products.

For each of the 13 validation targets we tabulate both unbiased RMSE (ubRMSE) and bias (see Table 3). Most interesting is the ubRMSE, which gives an indication of the overall data quality. This statistic includes the random error and errors resulting from a mismatch in variance (either seasonal or diurnal), but not the overall bias. Errors in spatial representation of the sites affect MW and TIR in the same way. In order to highlight the relative performance of the two satellite products with respect to the common benchmark, we compare their performance directly in Fig. 3.

Encouragingly, the multi-site average ubRMSE for the four European Fluxnet sites shows the MW-LST (2.3 K) to be only moderately higher then TIR-LST (2.1 K) for these frost-free and cloud-free observations. This is a positive result for MW-LST because of the extra processing needed to correct MW data for sensing depth differences with TIR. Both satellite products have a higher ubRMSE with the Ameriflux stations, but again the multi-site average ubRMSE for MW-LST (3.0 K) is only slightly higher than that for TIR-LST (2.8 K). Figure 2a compares the ubRMSE with in situ data directly for the two satellite technologies. The high correlation between the two methods is an indication that spurious effect of spatial representation of the site affects both methods to similar degrees. Of all the stations, MW-LST has a lower ubRMSE at 5 of the 13 stations and the only stations where we record more than 0.5 K difference in ubRMSE between TIR and MW-LST are Fluxnet station D (2.5 K for MW Vs. 1.7 K for TIR) and Ameriflux station I (3.5 K for MW Vs. 2.9 K for TIR) and J (3.4 K for MW Vs. 2.4 K for TIR) .

Because MW-LST is scaled directly to TIR-LST its bias is almost completely determined by the bias between TIR-LST and the site (Table 3 and Figure 2b.). The European Fluxnet sites fall within the MSG domain, and these data years were part of the data on which the scaling of MW-LST is trained (Holmes et al. 2015). The TIR LST in the domain of the Ameriflux sites is based on either GOES-East or GOES-West data, and is produced based on a 3.9/11 dual window technique rather than the more typical 11/12 micro split-window available to the MSG-based LSA SAF LST. Together this accounts for increased



scatter in bias between the two satellite products and the ground data when comparing MSG domain (circles) and GOES domain (squares).

Although the mean bias (Fig 2b) is almost identical, the bias in morning heating ($\Delta T$, Fig 2c) has more variation between the two satellite products. It is interesting that generally the satellite products overestimate $\Delta T$ compared to ground data: on

average they both overestimate the recorded heating at the stations by about 10 % .

### 3.2 Cloud Tolerance of Satellite LST

To test the stability of the MW-LST for increasing levels of cloudiness we took a closer look at the 4 sites in Europe and 3 in the US (site A-G). To isolate the effect of clouds on the agreement between satellite and ground observations, we first

remove structural differences by fitting a linear regression for each location, based on data with cloudiness below 20 % ($\Gamma$ – see section 2c). We then divide the data in 5 equal bins of increasing cloud coverage from 0 to 100 %. The RMSE and mean difference (bias) between the satellite data and the regression-corrected in situ data is then calculated for each 20 % cloud bin. The purpose is to test the assumption that MW-LST is tolerant to higher levels of cloud coverage.

Figure 3 shows the result of this analysis for location A-G (from left to right). For each location the data coverage (top row),

RMSE (middle row), and bias (bottom row) are displayed for the five 20% bins of cloudiness. First of all, the large increase in negative bias with increasing cloudiness for the TIR-LST product stands out. At all stations we see a clear negative bias in response to increasing cloudiness for TIR-LST, and the overall agreement between stations is striking. At 40-60% cloud cover, all but one station show a significant negative bias for TIR-LST. Above 60 % cloud cover all stations (where TIR-LST is still available, presumably due to failure of the cloud mask) show a negative bias of 2 K or more. This clearly shows

that for TIR-LST we have to accept the hypothesis that clouds affect the satellite LST estimate, even after a cloud mask is applied. It is well known that TIR observations are sensitive to clouds and that a failure to mask for cloud conditions will result in an underestimated LST (for land surface above freezing). Because of this systematic response to clouds, the bias metric by itself is a good indicator of the effect of cloud contamination in clear sky TIR-LST products. The symbols in the top row show the diminishing temporal sampling with increasing cloud cover. When we contrast this with the size of the bias

it is clear that the cloud mask as implemented in the LSA-SAF product (for sites A-D) is not sufficient at removing cloud artifacts. The GOES product (for sites E-G) appears to remove times with high cloud values more completely. Although investigating the efficacy of cloud masks for TIR techniques is not the purpose of this paper, it does help illustrate how cloud effects can be identified with these ground stations.

In clear contrast to the TIR-LST products, the response of MW-LST to increasing cloudiness is much more muted and not as

consistent across stations. Stations A, B, C, and E show no response in terms of bias and below 80 % cloud cover there is no station with a MW-LST bias of more than 1 K. One station shows a negative trend (D), and two stations show a positive trend (F and G). But only above 80 % cloudiness do these trends result in bias error of greater than 1 K. Because we see both positive and negative biases in the MW LST analysis we cannot rely solely only on the bias metric to assess the impact



of clouds. If there are cancelling biases affecting an individual station, this could suppress the bias. The increased retrieval error would still be reflected in an increased RMSE. However, the RMSE of MW LST changes minimally relative to its baseline value at 0-20 % cloudiness, and mirrors the size of the bias. This indicates that there is little potential for 'hidden' biases behind these numbers. For MW-LST we can therefore reject the hypothesis that clouds affect the satellite LST

estimate.

The MW-LST-Sparse product (Eq. 4) adopts the same scaling with TIR as the diurnal MW-LST but has much less sensitivity to the imposed shape of the diurnal model (DTC). For clear skies this distinction is negligible, as apparent from the almost identical values of ubRMSE shown in Table 3. The effect of the clear sky model is likely to be higher in days with cloudy or partial cloud covered sky. And although the sparse set only has 4-8 observations per day, it allows more

samples in days with complex temperature changes. Such days are removed from the MW-DTC product if no good match is found between the diurnal model and observations. We can therefore use the MW-LST-Sparse product to test for undue influence of the DTC model (and its related quality flags) on the relationships between LST errors and cloudiness. The response in bias of MW-LST-Sparse to increasing cloudiness is almost identical to the response of MW-LST for each station (see Fig. 3). In terms of RMSE the sparse set shows equal or higher values than the diurnally-continuous MW-LST product,

which is not surprising as it does not have the smoothing and quality control associated with the DTC model.

### 3.3 All-sky validation by satellite overpass time

The MW-LST record is a combination of different satellites. In the following analysis the validation results of the MW-LST product are broken down by time of day and satellite input record. All data pairs where the minimum temperature at the station stays above freezing are included in this analysis, regardless of cloud cover. It is interesting to compare these results

to the much simpler approach that uses a single linear regression model globally (Holmes et al., 2009). Table 4 lists RMSE, standard error of estimate (SEE, precision) and bias for the old and new approach. The statistics are aggregated for all locations as listed in Table 1. The mean scaling with TIR-LST results in a drop in bias for the MW-LST, reducing the average RMSE by 1 K. Part of this reduced RMSE results from the improved characterization of the amplitude of the diurnal cycle, which improves the slope at all times of day and accounts for 0.2 K of the improvement in RMSE. The impact on the

random error component (quantified here by SEE) is mixed – on average there is no change. Biggest improvements in all metrics are recorded for the forest locations (Sites C, E, and H).

### 4 Discussion

Considering all 8 locations used in the cloud analysis we see little to no response to clouds in terms of bias and RMSE for MW-LST and this allows us to reject the hypothesis that clouds negatively affect its accuracy. However, for three sites we do

find weak and opposing biases at higher cloud coverage which require an explanation. The wavelength of Ka-band (8 mm) is two orders of magnitude larger than a typical cloud droplet (10 µm). Therefore, any effect of clouds on MW-LST would





stem from changes in associated meteorological conditions like atmospheric vapor content and temperature profiles and their potential impact on Ka-band emission processes. According to the zero-order radiative transfer model, an increased atmospheric opacity (through increasing atmospheric water content) increases the weight of the atmospheric contribution to the satellite measured brightness temperature, relative to the top of vegetation emission. The sign and size of the effect of a

change in atmospheric opacity thus depends on the contrast between the atmospheric temperature and the land surface temperature times the effective emissivity. It is therefore possible that this could explain the site-to-site differences in bias as shown in Fig 3. Analyzing the overall effect of the atmosphere on biases in MW-LST will require more detailed atmospheric profile information coupled with a radiative transfer model.

Another possible explanation is that the positive biases recorded at locations F and G are related to scale differences between

the site and the 0.25 degree grid cell. Spatial heterogeneity in LST is likely more pronounced during clear-sky periods when spatially varying soil and vegetation wield a strong influence on the day time temperature gradients. During cloudy periods the temperature gradients are not as pronounced and more directly linked to the more uniform air temperature. If the mean temperature at the station is generally higher than the areal mean LST, and this bias diminishes with increasing cloudiness, this would be transformed through our clear-sky training into a positive bias for the satellite product at high cloudiness.

However, this effect would affect both MW and TIR to the same extent. We have tested this at locations with two stations in contrasting land cover types (A-C). What we found is that indeed it is possible to 'rotate' the bias response by changing the weights of the individual stations, and that this rotation affects both MW and TIR-LST. This effect of site representation can therefore explain the greater variation in response from station to station for locations where only one station was available (D-G).

**5 Conclusion**

In this paper, a recently developed satellite MW-LST product is compared to ground station data and satellite TIR-based LST products. The MW-LST was developed to complement TIR-LST with a coarser spatial resolution but at a higher temporal resolution. The higher temporal resolution of MW-LST is based on the assumption that MW has a relatively high tolerance to clouds, which allows for observations at times when no TIR observations are possible. This paper tests this

assumption by looking at the precision with respect to ground stations for increasing levels of estimated cloudiness. Our analysis is performed at the 0.25° spatial resolution as predicated by the MW-LST product. At this coarse spatial resolution, the overall unbiased RMSE between TIR-LST and ground stations during clear sky days is 2.1 K for the four locations in the MSG domain, and 2.8 K for the 9 locations in the GOES domain. For the same locations we find that the MW-LST is only slightly higher (+0.2 K for both domains).

With increasing cloudiness the RMSE increases significantly for TIR-LST, caused by a matching negative trend in bias that is seen at all seven locations. This demonstrated the known effect that clouds have on TIR estimates of LST. The fact that these trends are so apparent highlights the limitations of current cloud screening techniques as employed in TIR-LST



products that are in general use. In clear contrast to this we find a much more limited response in both RMSE and bias for MW-LST. Because of this we conclude that there is no significant direct impact of clouds on the accuracy of the MW-LST product. However, at three stations the size and sign of the response is such that further research is needed to identify the exact causes introducing error in MW-LST. By taking into account the atmospheric humidity and temperature profile further

analysis may investigate the extent to which this mixed response can be explained by atmospheric conditions associated with cloudiness. Alternatively, if a greater database were available of locations with flux tower sites in contrasting land covers this could be used to isolate the role of scale mismatch between station and the satellite product.

As an immediate outcome the result of this work highlights the utility of MW technology for cloud screening of TIR-LST. This is something that will be explored in future work. Ultimately, the goal is to find the best way of combining MW and

TIR technology for the estimation of LST from space.

**Data Availability**

Timeseries of MW-LST and TIR-LST covering the locations and time period of this paper are available upon request from the corresponding author. The global source data for MW-LST are publicly available. They are aggregated from several data centers and we would like to thank Goddard Earth Sciences (GES) Data and Information Services Center (DISC) for

archiving and distributing TRMM satellite as acquired by NASA's Earth-Sun System Division, National Snow and Ice Data Center for archiving and distributing Aqua-AMSR-E data, and NOAA's Comprehensive Large Array-data Stewardship System (CLASS) for dissemination of Defense Meteorological Satellite Program data. LSA SAF disseminates EUMETSAT products. This work further used data acquired by the FLUXNET community (fluxnet.ornl.gov) and in particular by the following networks: AmeriFlux and CarboEuropeIP.

**Acknowledgements**

This work was funded by NASA through the research grant "The Science of Terra and Aqua" (13-TERAQ13-0181). We would further like to thank Li Fang (NOAA) for preparation and interpretation of GOES LST.

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





**Table 1. List of ground validation targets.**

| Cluster ID, name | Site ID | Geographic location | Vegetation (IGBP) | W | $\varepsilon_L$ median (st.dev) | Clear sky absorbtion ($A_{clr}$) | Notes |
|---|---|---|---|---|---|---|---|
| A. DE-Thuringia | DE-Hai | 51.0792 ˚N, 10.453 ˚E | Deciduous Broadleaf Forest | .16 | 0.993 (0.001) | 0.33 | (a) |
| | DE-Geb | 51.1001 ˚N, 10.9143 ˚E | Croplands | .80 | 0.983 (0.004) | 0.31 | |
| B. DE-Dresden | DE-Tha | 50.9636˚N, 13.5669˚E | Evergreen needleleaf forest | .36 | 0.983 (0.005) | 0.26 | |
| | DE-Kli | 50.8928 ˚N, 13.52250 ˚E | Croplands | .61 | 0.993 (0.001) | 0.27 | |
| C. CZ-Billy Kris | CZ-BK1 | 49.50021˚N, 18.5368˚E | Evergreen needleleaf forest | .72 | 0.985 (0.001) | 0.29 | (a) |
| | CZ-BK2 | 49.4944˚N, 18.5429˚E | Grasslands | .23 | 0.985 (0.001) | 0.30 | |
| D. ES-LMa | ES-LMa | 39.9415˚N, 5.7734 ˚W | Savannas | .77 | 0.987 (0.001) | 0.25 | |
| E. US-Marys River | US-MRf | 44.6465˚N, 123.5515˚W | Evergreen needleleaf forest | 1.0 | 0.995 (0.000) | 0.27 | |
| F. US-Woodward | US-AR1 | 36.4267˚N, 99.42˚W | Grasslands | .98 | 0.993 (0.003) | 0.23 | |
| G. US-SGP Main | US-ARM | 36.6058 ˚N, 97.4888˚W | Croplands | 1.0 | 0.963 (0.011) | 0.23 | |
| H. US-Wind River | US-Wrc | 45.8205 ˚N, 121.9519 ˚W | Evergreen needleleaf forest | .99 | 0.993 (0.004) | 0.23 | |
| I. US-Santa Rita | US-SRC | 31.9083 ˚N, 110.8395 ˚W | Open Shrublands | .5 | 0.960 (0.008) | 0.21 | |
| | US-SRM | 31.8214 ˚N, 110.8661 ˚W | Woody Savannas | .5 | 0.983 (0.006) | 0.21 | |
| J. US-Audubon | US-Aud | 31.5907 ˚N, 110.5092 ˚W | Grasslands | .99 | 0.950 (0.003) | 0.20 | |
| K. US-Lucky Hills | US-Whs | 31.7438 ˚N, 110.0522 ˚W | Open Shrublands | 1.0 | 0.972 (0.017) | 0.19 | |
| L. US- | US-AR2 | 36.6358 ˚N, 99.5975 ˚W | Grasslands | 1.0 | 0.992 (0.003) | 0.26 | |
| M. US-Kansas | US-KFS | 39.0561 ˚N, 95.1907 ˚W | Grasslands | 1.0 | 0.945 (0.015) | 0.25 | |

a. Sites are located in neighboring grid cells.
Reference: C: (2011),



**Table 2. Percentage coverage for two LST products.**

| SAT Product | ALL | MSG Domain 2009-2011 | | | |
|---|---|---|---|---|---|
| | | All | Frost-Free Days | | |
| | | | All | Clear Sky | Clouded Sky |
| **TIR** | 36 | 42 | 47 | 94 | 14 |
| **MW** | 50 | 55 | 64 | 75 | 56 |





**Table 3. Summary of 'clear-sky' validation results.**

| European Fluxnet | Statistic | TIR-LST | MW-LST | MW-LST-Sparse |
|---|---|---|---|---|
| A: DE-Thuringia | ubRMSE | 2.2 | 2.4 | 2.4 |
| | BIAS | 0.4 | 1.2 | 1.1 |
| | N | 2050 | 2636 | 525 |
| B: DE-Dresden | ubRMSE | 1.6 | 1.7 | 1.7 |
| | BIAS | -0.6 | -0.7 | -0.9 |
| | N | 4157 | 4521 | 1245 |
| C: CZ-BK | ubRMSE | 2.6 | 2.6 | 2.6 |
| | BIAS | 2 | 2 | 1.9 |
| | N | 2787 | 2985 | 850 |
| D: ESLMa | ubRMSE | 1.7 | 2.5 | 2.6 |
| | BIAS | 0.4 | 0.8 | 0.7 |
| | N | 12055 | 9652 | 2560 |
| **Multi Site Average** | **ubRMSE** | **2.1** | **2.3** | **2.3** |
| | **\|BIAS\|** | **0.9** | **1.2** | **1.2** |

| Ameriflux | Statistic | TIR-LST | MW-LST | MW-LST-Sparse |
|---|---|---|---|---|
| E: US-Marys River | ubRMSE | 3.2 | 3 | 3.3 |
| | BIAS | 1.9 | 1.5 | 1.7 |
| | N | 2810 | 1991 | 756 |
| F: US-Woodward | ubRMSE | 3.1 | 3 | 2.8 |
| | BIAS | -0.6 | -0.5 | -0.5 |
| | N | 2925 | 10386 | 3811 |
| G: US-SGP Main | ubRMSE | 2.8 | 3.3 | 3 |
| | BIAS | 0.5 | 0.4 | 0.6 |
| | N | 2895 | 8362 | 3105 |
| H: US-Wind River | ubRMSE | 2.5 | 2.3 | 2.5 |
| | BIAS | -0.9 | -1.1 | -1.1 |
| | N | 4219 | 6915 | 1687 |
| I: US- Santa Rita | ubRMSE | 2.9 | 3.5 | 3.3 |
| | BIAS | -1.5 | -0.7 | -1.2 |
| | N | 8908 | 13478 | 5312 |
| J: US-Audubon | ubRMSE | 2.4 | 3.4 | 3.4 |
| | BIAS | -1.3 | -1.3 | -1.6 |
| | N | 3301 | 8345 | 3078 |
| K: US-Lucky Hills | ubRMSE | 3.1 | 3.5 | 3.5 |
| | BIAS | -0.7 | 0.3 | -0.1 |
| | N | 7870 | 10786 | 4545 |
| L: US-AR2 | ubRMSE | 3.2 | 2.6 | 2.4 |
| | BIAS | -0.5 | -0.6 | -0.6 |
| | N | 2033 | 8766 | 3124 |
| M: US-Kansas | ubRMSE | 2.3 | 2.1 | 2 |
| | BIAS | -0.2 | -0.1 | -0.4 |
| | N | 1454 | 4788 | 1627 |
| **Multi Site Average** | **ubRMSE** | **2.8** | **3** | **2.9** |
| | **\|BIAS\|** | **0.9** | **0.7** | **0.9** |



Table 4. Validation results broken down by MW satellite (all data, descending (D), ascending (A) path), aggregated for the 13 fluxnet sites.

| Satellite | Path | 2009 Linear regression | | | 2015 diurnal scaling | | | N |
|---|---|---|---|---|---|---|---|---|
| | | RMSE | SEE | BIAS | RMSE | SEE | BIAS | |
| AMSR-E | | 4.5 | 3 | 0.7 | 3.2 | 2.6 | 0.2 | 6227 |
| | D | 4.4 | 1.9 | 3 | 3.1 | 2.1 | -1.1 | 2817 |
| | A | 4.6 | 2.7 | -1.5 | 3.3 | 2.6 | 1.4 | 3410 |
| AMSR2 | | 4.4 | 2.4 | 1.6 | 3 | 1.9 | 0.8 | 843 |
| | D | 5 | 1.7 | 3.7 | 2.6 | 2 | -0.7 | 347 |
| | A | 3.8 | 1.6 | -0.3 | 3.2 | 1.6 | 1.9 | 478 |
| WindSat | | 3.5 | 2.3 | 0.8 | 3.1 | 2.5 | -0.6 | 3080 |
| | D | 3.3 | 2.1 | 1.4 | 3.1 | 2 | -0.2 | 1617 |
| | A | 3.7 | 1.9 | 0.2 | 3 | 2.1 | -0.9 | 1463 |
| SSM/I | | 3.8 | 2.4 | 0.9 | 3.1 | 2.6 | -0.3 | 5401 |
| | D | 3.5 | 2.3 | 1.1 | 3.1 | 2.2 | 0.2 | 2740 |
| | A | 4 | 2.1 | 0.8 | 3 | 2.3 | -0.8 | 2661 |
| **AVERAGE (All Sites)** | | **4.1** | **2.4** | **1.2** | **3.1** | **2.4** | **0.6** | |
| FOREST (Sites: C, E, H) | | 5.1 | 2.1 | 4.3 | 3 | 2 | 0.5 | |
| LOW VEGETATION | | 3.9 | 2.5 | 0.9 | 3.2 | 2.5 | 0.7 | |





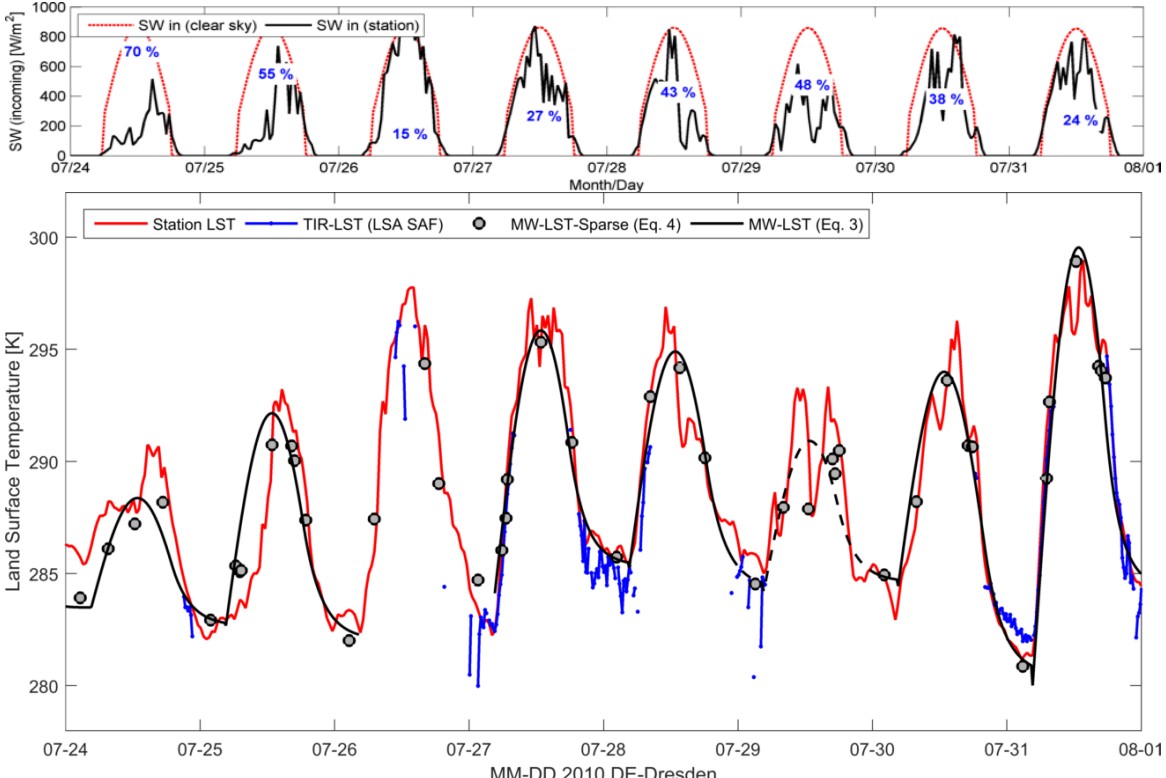

5 **Figure 1. An example of the available data at station B, showing 8-day time series of station measured shortwave incoming radiation ($S^{\downarrow}$, or SW in, top) and LST (bottom graph). In the top graph $S^{\downarrow}$ (black lines) is compared to the clear sky expected value, $S^{\downarrow}_{clear}$ (Eq. 7, red dash), to illustrate the computation of the cloud cover proxy ($A_{cloud}$, Eq. 8, values in blue text). In the lower graph the station LST is compared to the TIR and MW satellite LST products.**





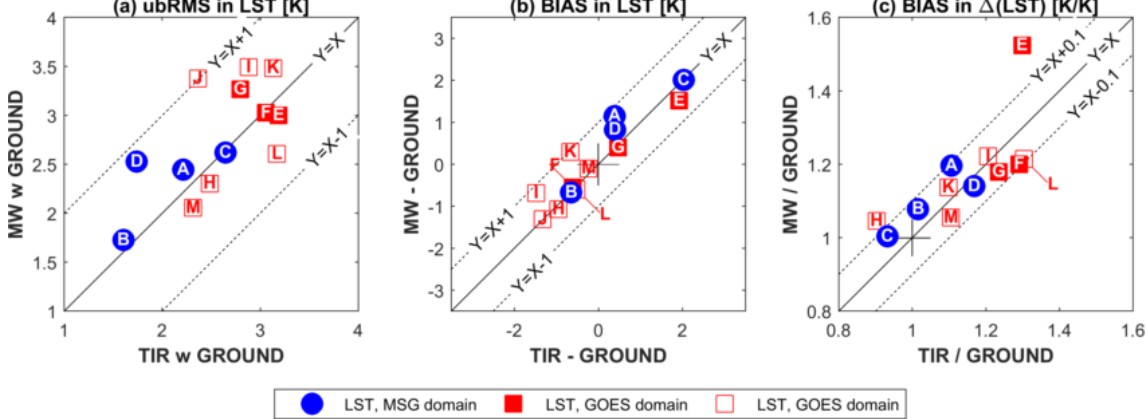

5 **Figure 2. Comparison between TIR-LST (X-axis) and MW-LST (Y-axis) in terms of their validation metrics with station LST for frost-free and cloud-free days. From left to right the three panels show (a) ubRMSE, (b) mean bias, and (c) bias in Δ(LST). Each marker represents the statistics as calculated for individual locations as identified by the letter (see Table 1 for definition). For the GOES domain the filled markers highlight the stations used in the cloud analysis. Black lines provide visual support and indicate targets (eg. 1:1 line, cross at zero bias (b,c)).**





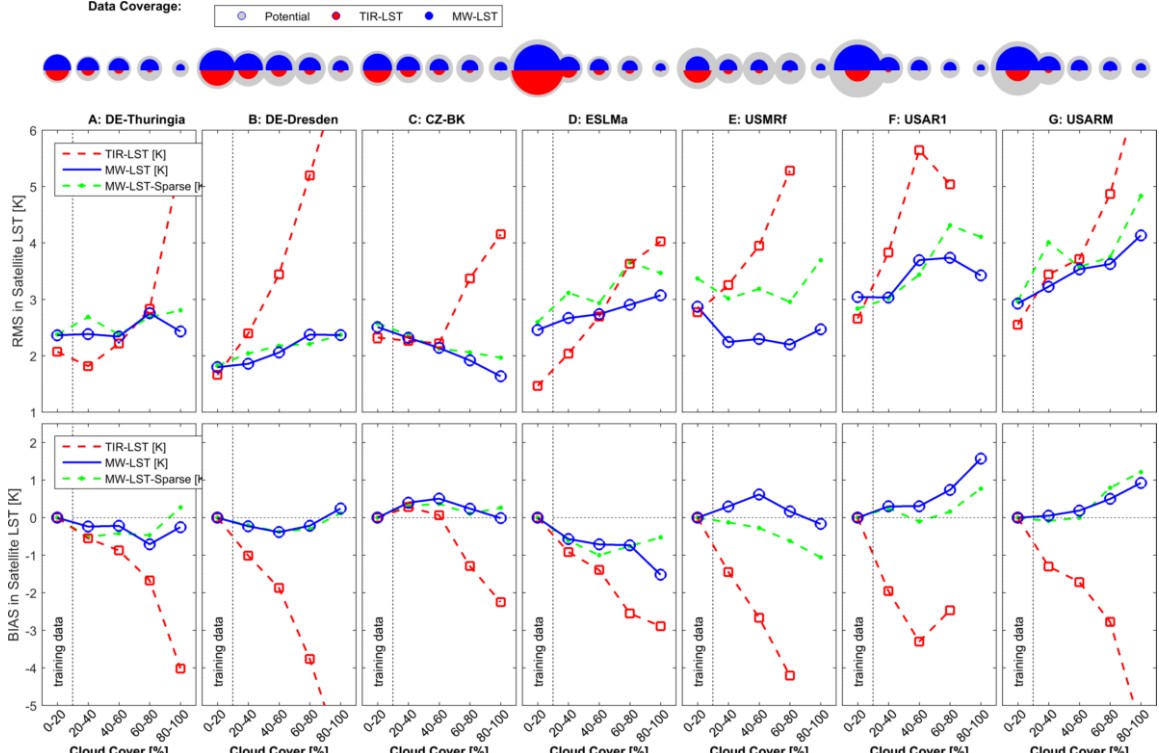

**Figure 3. RMSE and bias of satellite LST with regression corrected in situ data for five levels of cloud cover (Acloud, Eq. 8-10). From left to right are locations A-G (see Table 1 for site information). Results for TIR-LST (red) are contrasted with those for MW (blue). Markers indicate that more than 15 days with data were available for a particular cloud cover bin. Green dashed lines 5 indicate the results for MW-LST-Sparse. For each site and cloud interval the percentage coverage of the temporal record is depicted in the top row with half-rounds (MW: blue, TIR: red, and potential coverage in grey).**