# Peer review of "Cloud Tolerance of Remote Sensing Technologies to Measure Land Surface Temperature"

_Hydrology and Earth System Sciences, 2016_

## Referee Comment (RC1) · C. Prigent (Referee) · 19 May 2016

A methodology has been developed by Holmes et al. to produce LST diurnal cycle from the collection of available passive microwave observations on board satellite imagers. It is anchored to infrared LST estimates. In the present study, the authors evaluate the sensitivity of the microwave to the clouds, using in situ LST measurements along with in situ estimate of the cloudiness at the station locations. With the infrared LST estimates highly limited by clouds, there is today a strong interest in using microwave observations for LST retrievals to complement the IR. This study provides a valuable contribution in this field, with a quantification of the cloud impact on the microwave LSTs. The paper is well structured and well written. It deserves publication. However, before this study can be published, some issues need to be clarified.

[Figure]

Major points:

1) In this study, the microwave estimates of the LST diurnal cycle strongly rely upon the IR estimates. One expects the LST diurnal cycle to be impacted by clouds, even under partial cloud coverage along the day. The shape of the LST diurnal cycle (Holmes et al., 2015) will be affected if clouds are present at some time during the day. Can you clarify how these potential cloud effects are taken into account in the modelling / parameterization of the diurnal cycle? Two products are used in this study, the continuous MW-LST product and the MW-LST sparse product. It would be very interesting to see also results without any adjustments for the diurnal cycle. Would that be possible?

2) The number of MW-LST estimate is rather low as compared to what could be expected (50% with respect to 36% for the IR). Can you confirm that this is related to the need to have a very limited cloud coverage during the day, to anchor the microwave retrieval to the IR estimates? Would it be possible to relax this constraint, with modified assumption about the LST diurnal cycle (Holmes et al., 2015) under cloudy sky conditions?

3) The statistical metrics need to be better defined. The mean, the standard deviation, and the root mean square error are common statistical parameters that do not need to be explained. However, the ubRSME or the SEE need to be clarified. For instance, it was first understood that the SEE was the standard deviation, but from Table 4, it is obvious it is not the case (otherwise we would have RMSE2=STD2+BIAS2). The advice is to use the standard and commonly accepted statistical metrics (mean, std, rmse), to avoid any ambiguity, or define clearly the metrics used.

4) Clouds do affect the 37 GHz (p.10 l 31 sec. 4). The sentence 'Therefore. . .. Ka-band emission process' has to be removed or modified. Cloud particles do not scatter the radiation at 37 GHz (because of the small size of the particle with respect to the 37 GHz wavelength) but they do absorb and emit radiation. This is actually how the cloud water content is estimated over ocean from observations at 37 GHz. Over land, cloud

water retrieval is nevertheless very difficult, because of the lack of contrast between the cloud and the surface contribution to the signal (with an emissivity of the land surface close to 1 compared to the low emissivity of the ocean that provide a good contrast with the cloud emission).

Minor points:

- P.2 l.27. 'Well validated TIR-LST'. Not that sure... Our recent experiences show significant differences between TIR-LST products that are all supposed to be well validated...

- P.2. l.27-28. Could you provide more information about the sources of these biases?

- P.3. l23-24. How are the IR pixels handle within each 0.25° pixels? There are approximately 64 3km pixels in each 0.25° pixel. Do they all need to be clear to aggregate the data?

- P.4. l.5-6. Can you specify here if the inter-calibration is in TB or LST space. The information might be available in other papers, but it needs to be reminded here.

- P.5. eq. 4. It is not obvious here to realize how different is the MW-LST-Sparse from the 'raw' microwave LST. Can you comment on this point? Examples of the three MW-LST (continuous, sparse and 'raw') would help.

- P.7. l.10-11. Neighbouring daytime information taken for nighttime hours. That looks very uncertain. Information every 6 hours or every 3 hours as specified l9, but not used in l12-13? Did you try the analysis with only the daytime data?

- P.7. l.29. The 55% is after removal of the subfreezing temperatures? It will be good to give also the percentage only related to cloudiness.

- P.8 l.20. ThAn, not then

- P.9. l.1-2. Not that clear. MSG stations B,D,C are close to 1:1, as GOES stations J, H, M, G, E. Only K and J (GOES) are worse than D (MSG).
- P.9. l.29-30. How do the scaling of the MW-LST to the TIR-LST play a role here? It would be nice to see how the 'raw' MW-LSTs behave?

- P.11. Discussion. It would be nice to see the linear regression of Table 4 also discussed in Figure 3, to see if the MW-LST cloudiness dependence (even small) changes by not having the MW-LS somehow depending on the IR diurnal cycle.

- P.11 l.9. What is the spatial heterogeneity of the land surfaces for these two locations? That could be assed with the std of the IR LST within the large 0.25° pixel.

- Table1. Something missing for station L. US-? Wha is the meaning of column W?

- Table 2. CloudY not clouded.

- Fig1. When the situation is almost clear (i.e. on July 26 on the upper panel), several IR values were expected, with a good description of the diurnal cycle. This is not the case (lower panel). Can you explain?

- Fig1. For the A calculation, in the top figure, it looks like a full diurnal cycle of SW is available. In the text, only 6 hour information and/or 3 hour information are mentioned.

- Fig1. What is the meaning of the dashed black lines on July 29?

- Fig2. Avoid the use of the ubRMSE or define it. Can you comment on the differences between stations in terms of their soil moisture (or a proxy of the soil moisture if not available)? The amplitude of the diurnal cycle could actually be used.

- Fig3. The indication of the cloud conditions with the circles is not straightforward. It would be helpful to add more explanation in the caption.

- Fig3. The spatial standard deviation of the IR LST information at each station might help explain some of the differences between stations, in terms of surface heterogeneity.

Catherine Prigent and Carlos Jimenez

---

## Referee Comment (RC2) · Anonymous Referee #2 · 23 May 2016

The manuscript by Holmes et al. investigated the cloud tolerance of the microwave remote sensed land surface temperature (LST). They found that the clouds have no direct impact on the accuracy and bias of microwave-based LST. Since the thermal infrared based LST is highly restricted by clouds, the microwave based LST would provide complementary LST over cloudy conditions. The estimation of land surface heat fluxes and soil moisture would benefit a lot from the availability of LST over full sky conditions. Therefore, the current study is of great interest and worth for publication. In general, the manuscript is well organized and written. I only have a few comments that are listed below.

General comments:

1. Since there are other recent published studies on estimating LST with microwave

observations such as André et al., (2015), Prigent et al., (2016), I suggest the authors integrate these studies in the introduction section.

2. The study tested the hypothesis that microwave based LST is not sensitive to clouds over the FLUXNET tower scale. From the manuscript, it seems that the global scale microwave LST is already available. I suggest the authors to conduct a further study in the future on global analysis such as inter-comparison with other available LST products. Nevertheless, it would be nice if the authors could show the spatial pattern of the microwave LST and thermal LST. In addition, the current study focuses on the time period 2009-2011. For hydrological applications, this period is very limited. Does the developed microwave LST cover longer period? If not, do you have plans to extend it to long time period? I did not find detailed description on the global microwave product.

3. The diurnal temperature cycle (DTC) of microwave LST is scaled to match that of thermal LST. Therefore, the diurnal microwave LST depends on the thermal LST to some extent. I am wondering how much influence would this scaling bring to the diurnal microwave LST. In other words, I suggest the authors show the results before scaling the DTC to match thermal LST.

Specific comments:

1. Page 2 Line2: specify the name of the radiometer that deliver 2 km spatial resolution.

2. Page 4 Line3: List the names of the 6 staellites/sensors.

3. Page 5 Line7: It would be nice if the used FLUXNET towers are shown in a global map.

4. The statistic metric R (correlation coefficient) should also be calculated except for ubRMS and BIAS.

References: André, C., et al. "Land surface temperature retrieval over circumpolar Arctic using SSM/I–SSMIS and MODIS data." Remote Sensing of Environment 162 (2015): 1-10.

Prigent, C., C. Jimenez, and F. Aires. "Towards "all weather", long record, and re-al‐time land surface temperature retrievals from microwave satellite observations." Journal of Geophysical Research: Atmospheres (2016).

---

## Author Comment (AC1) · 10 Jun 2016

We would like to thank the referees for their insightful and thorough review of the paper. All the referee's comments are addressed in the following response. The manuscript is being revised to accommodate these changes.

1) In this study, the microwave estimates of the LST diurnal cycle strongly rely upon the IR estimates. One expects the LST diurnal cycle to be impacted by clouds, even under partial cloud coverage along the day. The shape of the LST diurnal cycle (Holmes et al., 2015) will be affected if clouds are present at some time during the day. Can you clarify how these potential cloud effects are taken into account in the modeling / parameterization of the diurnal cycle? Two products are used in this study, the continuous MW-LST product and the MW-LST sparse product. It would be very interesting to see

also results without any adjustments for the diurnal cycle. Would that be possible?

Reply: The influence of IR on the MW-LST estimates is solely through a time-constant linear scaling of the diurnal minimum, amplitude and timing. Therefore, this approach should not impact the presented results pertaining to the cloud effect on MW LST. But indeed, the diurnal cycle imposed on the MW-LST reflects an idealized clear sky situation. The result is that especially during partly cloudy days the fitted diurnal may not reflect the actual LST at the ground. Examples of this are apparent in Fig. 1 of the manuscript (eg June 24, 25, 28). However, the diurnal amplitude is determined based on a least-squares fit between diurnal model and daytime brightness temperature observations. This means that the overall diurnal increase in temperature is based on the observations, but not the hour-to-hour shape. As to the second part of the question; the difference between sparse observations and the diurnal temperature model is preserved in MW-LST-Sparse. In Fig 1. the effect of this is clear - MW-LST-Sparse corresponds more closely to the ground observations of LST than the continuous MW-LST. It is therefore used in this paper to present results with minimal effect of the diurnal model.

2) The number of MW-LST estimate is rather low as compared to what could be expected (50% with respect to 36% for the IR). Can you confirm that this is related to the need to have a very limited cloud coverage during the day, to anchor the microwave retrieval to the IR estimates? Would it be possible to relax this constraint, with modified assumption about the LST diurnal cycle (Holmes et al., 2015) under cloudy sky conditions?

Reply: The first column of Table 2 lists the overall data availability. The 50 % coverage for MW is indeed lower than what one would expect but this is not only due to cloud constraints. First of all the number is lower due the half year interval between AMSR-E and AMSR2 in 2012 (column 2 focuses on 2009-2011 for this reason). Secondly, the presented MW-LST excludes temperatures below freezing, and days where the temperature dips below freezing are excluded from the analysis. So column 4, with 64

% coverage of MW-LST in frost free days is closer to the general expectation of this method. The primary limitation in these conditions is the requirement of an observation near the daily maximum to constrain uncertainty of the diurnal amplitude. This effectively limits the method in its current implementation to days with an AMSR-E or AMSR2 ascending overpass: 2 out of 3 days. Incorporation of FenYun 3b could bring the coverage closer to daily.

3) The statistical metrics need to be better defined. The mean, the standard deviation, and the root mean square error are common statistical parameters that do not need to be explained. However, the ubRSME or the SEE need to be clarified. For instance, it was first understood that the SEE was the standard deviation, but from Table 4, it is obvious it is not the case (otherwise we would have RMSE2=STD2+BIAS2). The advice is to use the standard and commonly accepted statistical metrics (mean, std, rmse), to avoid any ambiguity, or define clearly the metrics used.

Reply: We added a short section on Statistical metrics (2.4) that defines bias, RMSE, ubRMSE and SEE.

4) Clouds do affect the 37 GHz (p.10 l 31 sec. 4). The sentence 'Therefore Ka-band emission process' has to be removed or modified. Cloud particles do not scatter the radiation at 37 GHz (because of the small size of the particle with respect to the 37 GHz wavelength) but they do absorb and emit radiation. This is actually how the cloud water content is estimated over ocean from observations at 37 GHz. Over land, cloud water retrieval is nevertheless very difficult, because of the lack of contrast between the cloud and the surface contribution to the signal (with an emissivity of the land surface close to 1 compared to the low emissivity of the ocean that provide a good contrast with the cloud emission).

Reply: Thanks for this insightful comment on the theoretical bases of the pathway through which clouds would affect Ka-band emission. We changed the sentence to: "Therefore, clouds do not scatter the 37 GHz radiation coming from the surface. They

do however absorb and emit radiation themselves. The effect of clouds on MW-LST would thus be moderated by associated meteorological conditions like atmospheric vapor content and temperature profiles."

Minor points:

- P.2 l.27. 'Well validated TIR-LST'. Not that sure: Our recent experiences show significant differences between TIR-LST products that are all supposed to be well validated: Reply. Point well taken. In the revised paper we focus the sentence on the geostationary aspect of the TIR-LST rather than its level of validation.

- P.2. l.27-28. Could you provide more information about the sources of these biases? Reply: This scaling was able to account for biases in characteristics of the diurnal temperature cycle (DTC) related to Ka-band emissivity, sensing depth, and atmospheric effects (Holmes et al. 2015).

- P.3. l23-24. How are the IR pixels handle within each 0.25 pixels? There are approximately 64 3km pixels in each 0.25 pixel. Do they all need to be clear to aggregate the data? Reply. If two-thirds of the 3-km observations are masked than the sample average is rejected for that location and time.

- P.4. l.5-6. Can you specify here if the inter-calibration is in TB or LST space. The information might be available in other papers, but it needs to be reminded here. Reply. We clarified in the text that the intercalibration is based on the brightness temperatures.

- P.5. eq. 4. It is not obvious here to realize how different is the MW-LST-Sparse from the 'raw' microwave LST. Can you comment on this point? Examples of the three MW-LST (continuous, sparse and 'raw') would help. Reply. We added the following explanation below Eq. 4. "In MW-LST-Sparse the impact of the DTC3 model is limited to providing the minimum and amplitude of the diurnal. The differences between the observations and the diurnal model at the actual observation time are preserved. The difference between the continuous MW-LST and the MW_LST-Sparse is illustrated in

Fig. 1 (lower panel). "

- P.7. l.10-11. Neighbouring daytime information taken for nighttime hours. That looks very uncertain. Information every 6 hours or every 3 hours as specified l9, but not used in l12-13? Did you try the analysis with only the daytime data? Reply. We did do the analysis with only day-time data but this makes the results more influenced by biases in the diurnal amplitude. By using this estimate of cloudiness for night-time hours allows to use the night-time data.

- P.7. l.29. The 55% is after removal of the subfreezing temperatures? It will be good to give also the percentage only related to cloudiness. Reply. The final two collumns in Table 2 list the coverage for clear and cloudy sky. From this it can be seen that for TIR, the cloudy sky coverage is 15 % of clear sky coverage. For MW the cloudy sky coverage is 75% of the clear sky value. The 25 % gap between cloudy and clear sky coverage for MW is related to the spatial standard deviation filter applied to AMSR-E, TMI, and Windsat and this is thought to capture effects of active precipitation. Furthermore days where the difference between the sparse observations and the diurnal model are too large are removed.

- P.8 l.20. ThAn, not then Reply. It has been changed.

- P.9. l.1-2. Not that clear. MSG stations B,D,C are close to 1:1, as GOES stations J, H, M, G, E. Only K and J (GOES) are worse than D (MSG). Reply. Indeed. This explanation of increased scatter pertains to the ubRMSE.

P.9. l.29-30. How do the scaling of the MW-LST to the TIR-LST play a role here? It would be nice to see how the 'raw' MW-LSTs behave? Reply. Matching the diurnal amplitude has a big effect on the efficacy of this method to detect cloud effects. If there is big mismatch between diurnal amplitude of site and satellite, then any variation in diurnal sampling between cloud bins will result in a bias that is superimposed on the cloud response signal. With a longer data record such spurious effects may be mitigated but for now we think it's best to focus on the data sets in 'LST' space.

- P.11. Discussion. It would be nice to see the linear regression of Table 4 also discussed in Figure 3, to see if the MW-LST cloudiness dependence (even small) changes by not having the MW-LS somehow depending on the IR diurnal cycle. Reply. Just as in our response to the comment above, the large influence of the bias in diurnal cycle with respect to the ground data makes it very complicated to analyze datasets with different diurnal biases. The 2009 regression does not account for the diurnal amplitude bias and will therefore underestimate the diurnal cycle in LST. It would be hard to disentangle the effect of changing sampling biases from actual cloud induces biases.

- P.11 l.9. What is the spatial heterogeneity of the land surfaces for these two locations? That could be assessed with the std of the IR LST within the large 0.25 pixel. Reply. The spatial standard deviation of IR LST could indeed be a good indicator for spatial heterogeneity during clear sky periods, if one could make sure to avoid cloud artefacts in the data. However, it is not possible to get the same information for cloud covered scenes.

- Table1. Something missing for station L. US-? Wha is the meaning of column W? Reply. The table is amended to clarify these meanings.

- Table 2. CloudY not clouded. Reply. It has been changed.

- Fig1. When the situation is almost clear (i.e. on July 26 on the upper panel), several IR values were expected, with a good description of the diurnal cycle. This is not the case (lower panel). Can you explain? Reply. We don't know for sure the reason behind these data gaps. As with any cloud mask, there could be false negatives as well as false positives. It may also be that significant cloud cover affected the 0.25 degree grid box but passed the station by.

- Fig1. For the A calculation, in the top figure, it looks like a full diurnal cycle of SW is available. In the text, only 6 hour information and/or 3 hour information are mentioned. Reply. Shortwave radiation is reported at an hourly or half-hourly interval. The measure of cloudiness is calculated based on 3-hr intervals in order to provide a more timeaveraged metric that might be more representative of the 0.25 degree spatial average.

- Fig1. What is the meaning of the dashed black lines on July 29? Reply. The dashed line represents a case where the MW-LST was calculated but rejected based on a large difference between sparse observations and the diurnal fit. This wasn't explained in the text and we added it to the caption.

- Fig2. Avoid the use of the ubRMSE or define it. Reply: We defined the use of ubRMSE more clearly in Section 2.4.

- Fig2. Can you comment on the differences between stations in terms of their soil moisture (or a proxy of the soil moisture if not available)? The amplitude of the diurnal cycle could actually be used. Reply. This is an interesting observation. Locations D, I, J, K, L all have dry conditions with low vegetation. In Fig 2a all but (L) have a 0.5 to 1 k higher ubRMS for MW than for TIR. When there is less vegetation, the MW transmissivity is higher and the influence of soil emissivity on the observed Ka-band brightness temperature becomes larger. Small changes in soil moisture can affect the soil emissivity and will result in biases for the MW-LST when a constant emissivity is assumed (as in current implementation). This points to possible improvements when the scaling to TIR is performed at shorter window lengths, perhaps in 3-month moving windows. We added this discussion in the description of the Fig 2a in Section 3.1.

- Fig3. The indication of the cloud conditions with the circles is not straightforward. It would be helpful to add more explanation in the caption. Reply. We added explanation in the caption: "For each site and cloud interval the percentage coverage of the temporal record is depicted in the top row with half-rounds in proportion to the number of data pairs. The potential number of data pairs (grey) refers to the number of in situ data points for each cloud bin. The actual number of data pairs is superimposed on this for MW (blue) and TIR (red). "

- Fig3. The spatial standard deviation of the IR LST information at each station might help explain some of the differences between stations, in terms of surface heterogeneity. Reply. This would indeed be good additional information to assess the spatial complexity of LST for each station and might be something to take into account in future studies.

---

## Author Comment (AC2) · 10 Jun 2016

We would like to thank the referee for their thorough review of the paper. All the referee's comments (included below in italics) are addressed in the following response. The manuscript is being revised to accommodate these changes.

Reply to General comments:

1. Since there are other recent published studies on estimating LST with microwave observations such as André et al., (2015), Prigent et al., (2016), I suggest the authors integrate these studies in the introduction section.

Reply. The recent papers of André et al., (2015), Prigent et al., (2016) are now referenced in the introduction.

[Figure]

2. The study tested the hypothesis that microwave based LST is not sensitive to clouds over the FLUXNET tower scale. From the manuscript, it seems that the global scale microwave LST is already available. I suggest the authors to conduct a further study in the future on global analysis such as inter-comparison with other available LST products. Nevertheless, it would be nice if the authors could show the spatial pattern of the microwave LST and thermal LST. In addition, the current study focuses on the time period 2009-2011. For hydrological applications, this period is very limited. Does the developed microwave LST cover longer period? If not, do you have plans to extend it to long time period? I did not find detailed description on the global microwave product.

Reply. This study is indeed focused on testing the cloud tolerance of MW-LST by using detailed all-weather records of ground observations. At larger scales a detailed analysis of the diurnal characteristics and an assessment of random error over Europe, Africa and the middle-East was described in Holmes et al. (2015). In terms of global comparisons with other LST products and for longer timescales, we intend to make the dataset available to facilitate these studies in collaboration with other investigators.

3. The diurnal temperature cycle (DTC) of microwave LST is scaled to match that of thermal LST. Therefore, the diurnal microwave LST depends on the thermal LST to some extent. I am wondering how much influence would this scaling bring to the diurnal microwave LST. In other words, I suggest the authors show the results before scaling the DTC to match thermal LST.

Reply. The scaling of MW TB to TIR LST is done at per pixel basis and uses three parameters that are held constant over time: difference in diurnal timing, amplitude and daily minimum. These constant scaling parameters cannot affect the potential response to time-variant impact of clouds. If there are effects on the analysis than that would be through physical thresholds imposed (freezing point), or diurnal biases with the ground data that are sampled different with different cloud cover bins. This type of effect would add noise to our analysis but not fundamentally change the conclusions. Moreover, the fact that MW and TIR have such a different response to clouds is

testimony to the independence of the MW-LST.

Reply to Specific comments:

1. Page 2 Line2: specify the name of the radiometer that deliver 2 km spatial resolution. Reply: The Spinning Enhanced Visible and Infrared Imager (SEVIRI) is now given as an example with 3 km resolution.

2. Page 4 Line3: List the names of the 6 satellites/sensors. Reply: These satellites include the Advanced Microwave Scanning Radiometer on EOS (AMSR-E) to October 2011 and its follow on AMSR2 from July 2012. Several platforms of the Special Sensor Microwave and Imager (SSM/I), the Tropical Rainfall Measurement Mission (TRMM) Microwave Imager (TMI), and Coriolus-WindSat.

3. Page 5 Line7: It would be nice if the used FLUXNET towers are shown in a global map. Reply: We include two maps of the Fluxnet tower locations here (Figs 1 and 2, below), but we feel it is not of sufficient added value to the information in Table 1 to add to the manuscript.

4. The statistic metric R (correlation coefficient) should also be calculated except for ubRMS and BIAS. Reply. The pearson correlation coefficient is not as discerning in the case of temperature when there is a strong seasonality that dominates the correlation. For the locations in this study the R values are between 0.94 and 0.99. MW and TIR perform very similar with only a few stations with a 0.01 lower R for MW. Only station J has a markedly lower R for MW-LST (0.91) Vs TIR (0.95).

Work cited: Holmes, Thomas R. H., Wade T. Crow, Christopher R. Hain, Martha Anderson, and William P. Kustas. 2015. "Diurnal Temperature Cycle as Observed by Thermal Infrared and Microwave Radiometers." Remote Sensing of Environment 158C: 110–25. doi:10.1016/j.rse.2014.10.031.
* * *
[Figure]

Fig. 1. Map of European locations of fluxnet towers used in this study.

**Fig. 2.** Map of US locations of fluxnet towers used in this study.